# Preparation and Study of Composite Materials of the NiAl-Cr-Mo-Nanoparticles (ZrO_2_, MgAl_2_O_4_) System

**DOI:** 10.3390/ma15175822

**Published:** 2022-08-24

**Authors:** Leonid Agureev, Valeriy Kostikov, Svetlana Savushkina, Zhanna Eremeeva, Maxim Lyakhovetsky

**Affiliations:** 1Department of Nanotechnology, Keldysh Research Center, Moscow 125438, Russia; 2National University of Science and Technology MISIS, Moscow 119049, Russia; 3Moscow Aviation Institute, National Research University, Moscow 125993, Russia

**Keywords:** composite material, intermetallic compound, spark plasma sintering, nanoparticles, Young’s module, bending strength

## Abstract

Materials based on the NiAl-Cr-Mo system with zirconium oxide or aluminum-magnesium spinel nanoparticle small additions were obtained by spark plasma sintering. Thermodynamic modeling was carried out to predict the phase formation in the NiAl-Cr-Mo system and its change depending on temperature, considering the presence of a small amount of carbon in the system. The phase composition and microstructure of materials were studied. NiAl (B2) and CrMo phases were found in the sintered samples. Bending strength measurements at different temperatures shows that nanoparticles of insoluble additives lead to an increase in bending strength, especially at high temperatures. A fractographic analysis of the sample’s fractures shows their hybrid nature and intercrystalline fracture, which is confirmed by the clearly visible matrix grains similar to cleavage. The maximum strength at 700 °C (475 MPa) was found for material with the addition of 0.1 wt.% zirconium oxide nanoparticles. In the study of internal friction, typical peaks of a nickel-aluminum alloy were found in the temperature ranges of 150–200 °C and 350–400 °C.

## 1. Introduction

Nickel-aluminum materials are promising for different parts of the manufacture of power plants and engines operating at high temperatures and in aggressive media. Compared to nickel alloys, the NiAl intermetallic compound has a high melting point (1911 K) and low density, a high heat resistance up to 1273 K, and good thermal conductivity (70–80 W/(m·K) at temperature range 25–1127 K) [1,2,3,4]. The NiAl compound has a density of 5.7–6.3 g/cm^3^ and its operating temperature can reach 1000–1100 °C. In addition, the material has good high-temperature strength, corrosion resistance, and creep resistance [5,6,7,8,9]. However, nickel-aluminum is non-plastic and brittle at room temperature, which reduces the possibility of its application [10,11,12,13]. There are several ways to improve the characteristics of this material using additives of transition, rare-earth, and noble metals [14,15,16,17,18,19,20]. The development of so-called eutectic alloys based on NiAl containing Cr, Mo and other elements improves its functional properties [12]. Eutectic materials in the NiAl-Cr-Mo system have high thermodynamic stability [14,15,16,17,21].

Intermetallic compounds have a high potential for use in composites with high heat resistance and high-temperature strength for rocket and space technology, power engineering and the aircraft industry [22,23]. In addition to its attractiveness as an engineering material, NiAl has such important properties as a simple crystalline structure, a high-ordered crystalline lattice, stability over a wide range of concentrations, a reversible shape memory effect, and anisotropic elastic and plastic behavior. Casting, rapid crystallization, powder metallurgy and mechanical synthesis are used to obtain materials based on NiAl. Limited plasticity and impact strength at room temperature make them difficult to manufacture using traditional processing methods. Meanwhile, due to the high melting point of NiAl and the requirements for vacuum facilities, casting methods have significant production costs [24]. The application of the powder metallurgy method is complicated by the presence of a stable surface oxide layer, and therefore, most aluminides are difficult to sinter to maximum density without pressure use [25]. Powder metallurgy makes it possible to economically create new materials from mixtures of mechanically activated metals and intermetallic compounds. In particular, the use of spark plasma sintering facilitates their rapid production with high density. According to [26], the chemical elements added to nickel alloys affect its structure and properties in different ways; for example, Al, Cr, and Mo increase strength and corrosion resistance. 

One of the reasons for the use of intermetallic materials, including NiAl, is that the chemical bonds in them consist of combinations of partially ionic or covalent bonds, which have a high strength compared to weak metallic bonds. CrMo alloys consisting of a solid solution are characterized by high heat resistance and corrosion resistance, which makes them promising as additions to NiAl-based materials. Moreover, the addition of small amounts of insoluble nanoparticles to a metal or alloy matrix leads to an increase in the strength characteristics and creep resistance, as well as to an improvement in a number of functional properties [27,28,29]. This paper describes the study of materials based on the system (at.%) NiAl-13Cr-13Mo (marked as “M”) with small additions of zirconium oxide (0.01–0.1 wt.%) or aluminum magnesium spinel (0.01–0.2 wt.%) nanoparticles, formed by spark plasma sintering of mechanically activated powders. The effect of small additions of nanoparticles on mechanical properties has not yet been fully explained. In a number of previous works, it has been found that in the case of a soft matrix, for example, aluminum [29,30], small additions of high-modulus insoluble in the matrix nanoparticles provide not only an increase in strength, but also in elastic modulus. Moreover, for a number of cases, such dependencies of properties on the concentration of additives of nanoparticles have an extreme character. This behavior phenomenon is explained by the formation of interfacial hardening zones around nanoparticles, which, according to the Obraztsov-Lurie-Belov theory, have a gradient change in the elastic modulus or strength from the boundary with the nanoparticle to the matrix [31]. In this work, we investigated the effect of small amounts of ZrO_2_ and MgAl_2_O_4_ nanoparticles, which have Young’s modulus of 577 GPa [32] and 275 GPa [33], respectively, on mechanical properties of the NiAl-CrMo matrix. Thermodynamic modeling was carried out in the Terra^®^ software environment [34] to predict the phase formation in the NiAl-Cr-Mo system, as well as with a small amount of carbon that enters the matrix from graphite molds during spark plasma sintering [35,36], considering the formation of a regular Cr-Mo solid solution. A fractographic analysis of the obtained samples was carried out, as well as studies of the features of the temperature dependence of internal friction.

## 2. Materials and Methods

The technology of nickel-aluminum matrix composites manufacturing included the preparation and mixing of initial powders NiAl (99.5%, 45 μm, PN70Yu30), JSC “Polema” (Tula, Russia), Cr (98.5%, 150 μm, PH1S, JSC “Polema” (Tula, Russia)), Mo (99.5%, 5 μm, MPCh, LLC “Atomspecsplav”, Saint Petersburg, Russia), with small amounts of MgAl_2_O_4_ nanoparticles (specific surface area—120 m^2^/g, average diameter—15 nm) or ZrO_2_-8Y_2_O_3_ nanoparticles (specific surface area—25 m^2^/g, average diameter—40 nm).

The sieving of the matrix powder was carried out in a special vibratory sieve to remove contaminants and large particles aggregates (with size more than 20 μm). Mechanical activation was carried out in the planetary mill «Activator—2SL». Mixing was carried out in steel cups in argon with steel balls. The ratio of powder to balls was 1:5, and the mixing time was 20 min. The disk rotation frequency was 33 Hz. After mixing in a planetary mill, the powders were extracted into the air. Then, isopropyl alcohol was added to them in the amount 1:5 of powder to liquid. Mixing was carried out with an overhead stirrer at a speed of 300 to 450 rpm to prevent foaming formation. During 5 min of stirring, the jar with the powder was in an ultrasonic bath at a frequency of 20 kHz. While stirring, the addition of zirconium oxide or aluminum-magnesium spinel nanoparticles pre-dispersed in isopropyl alcohol was carried out in various amounts (in the range of 0.01–0.2 wt.%). After finish of mixing, the jar was left to air dry for 24 h under a hood. Then the powder was extracted and pressed into graphite molds with a diameter of 30 mm. Sintering was carried out by the spark plasma sintering (SPS) using FCT System GmbH equipment at a pressure of 40 MPa and a temperature of 1300 °C in argon for 30 min with 130°/min heating rate to sintering temperature. There was no additional heat treatment of the sintered samples.

The study of the matrix “M” powder sample was carried out by thermal analysis using differential scanning calorimetry (DSC) and thermal gravimetric analysis (TGA) by NETZSCH^©^ STA449F 1 Jupiter thermal analyzer in alumina crucible in the temperature range from 25 °C to 1500 °C with heating rate 10°/min in an inert atmosphere (argon).

The morphology and composition of the materials were studied by scanning electron microscope (SEM) Quanta 600 with TRIDENT XM4 energy-dispersive X-ray spectroscopy (EDX) equipment. X-ray phase analysis (XRD) was performed in β-filtered cobalt radiation by a horizontal 2θ-θ X-ray diffractometer HZG-4. The phases were identified using the ICDD PDF-2 database. The ultimate bending strength was determined by the three-point method on a Test Systems-VacEto universal mechanical testing machine. Young’s modulus and internal friction were measured by the ultrasonic method on the MUZA^®^ device.

Thermodynamic modeling was carried out using the Terra^®^ software v. 6.2 by Bauman Moscow State Technical University, Moscow, Russia [34,37], which helps to study the chemistry of transformations in a wide range of compositions, temperatures and pressures. Equilibrium compositions depending on temperature, total enthalpy, entropy and equilibrium heat capacity of the system were found. The correct calculations of the system equilibrium imply the availability of data on the existing phases, their transformations, and thermodynamic properties. Terra^®^ software allows modeling of partially non-equilibrium processes using the following methods [38]:exclusion from the calculation of phases that do not have enough time to form. For example, during SPS, rapid heating and rapid quenching of the material occur, and the formation of a number of intermetallic compounds or solid solutions can be excluded;‘freezing’ the absolute content of some reagents, for example, with high phase stability, as well as low reactivity or short process times.

In addition, a number of assumptions was made during the calculations: the systems are closed and isolated in a state of internal thermodynamic equilibrium, surface interfaces at the phase boundaries are not considered, substances in different phases are considered different components.

The results of thermodynamic modeling were compared with experimental data from [39,40,41,42,43,44].

## 3. Results

### 3.1. Study of the Matrix Powder and Nanoparticles

Figure 1 shows the general view of the used nanoparticles to modify the intermetallic matrix. The zirconium oxide and spinel nanoparticles previously has spherical or near-spherical form (Figure 1a,b).

Figure 2 shows the microstructure of the matrix powders NiAl, Cr, Mo (without the addition of nanoparticles) after mixing the initial powders in a ball mill for 2 days. Matrix particles have a shape close to spherical, with a diameter from 5 to 40 μm. The average diameter of mixed powder particles before milling was 40 μm, and after milling—25 μm.

Thermal analysis of the sample shows that it is stable up to 1290 °C because there are not any peaks on the DSC curve. In the temperature range from 1290 °C to about 1350 °C, the DSC curve shows a double overlapping endothermic process with peaks at temperatures of 1300 °C and 1335 °C (Figure 3a). Further, at a temperature of 1450 °C, the beginning of another endothermic process is observed. In this case, there is practically no change in mass. A slight mass increase of less than 0.4% approximately corresponds to the observational error. The endothermic peak is associated with the onset of early melting. XRD analysis after DSC shows formation of Al_5_Mo intermetallic compound (Figure 3b). In addition, this could be due to the dissolution of the CrMo phase in the NiAl matrix.

Figure 4 shows the results of thermogravimetric analysis of the matrix powder. Weight gain starts at 700 °C and reaches 0.36% at 1500 °C due to slight oxidation in the presence of impurity oxygen.

### 3.2. Results of Thermodynamic Modeling

Thermodynamic calculations with varying initial parameters help to reduce the number of experiments to find the optimal composition and technological parameters of new materials formation process. This is especially important for non-equilibrium or quasi-equilibrium processes occurring under conditions of rapid temperature change. Data on the intermetallic compounds and various phases formation are taken from a number of verifiable sources [39,45,46,47,48,49,50,51,52,53,54,55,56], as well as form the IVTANTERMO^®^ database, the US National Bureau of Standards [57,58]. Due to the lack of research and the values of changes in thermodynamic properties, the standard entropy, standard enthalpy, density, and coefficients for the heat capacity polynomial are calculated according to the well-known methods [59,60,61,62,63]. In the database of thermodynamic substances used in the Terra^®^ software, properties are expressed in different ways. In thermodynamic modeling the concept of total enthalpy *I* is also used [64,65]. It is necessary to know the formula of the compound, as well as the values of the enthalpy of formation $\Delta H_{f}^{0}$, standard enthalpy $\Delta H_{298}^{0}$ and entropy $S_{298}^{0}$. All necessary initial data for calculations are taken from [66,67,68]. The results of thermodynamic modeling are shown in Figure 5.

### 3.3. XRD Analysis

The diffraction patterns of the matrix sintered samples («M», NiAl-13Cr-13Mo, at.%) with MgAl_2_O_4_ or ZrO_2_ nanoparticles additions are shown in Figure 6. NiAl B2, CrMo phases were found in the sintered samples.

Due to the closeness and high intensity of the main phases peaks and the peaks of intermetallic phases that could be formed during sintering (for example, related to the Al-Mo system) they could not be identified.

### 3.4. Elemental Composition

The map of the elements distribution in the M-0.01 MgAl_2_O_4_ composite structure is shown in Figure 7. The light areas in the figures represent clusters of molybdenum particles, which also contain chromium and nickel. Such regions were observed throughout the material. The formation of these areas is associated with the complexity of the distribution of molybdenum small particles among large particles of the nickel-aluminum matrix during stirring in a ball mill. However, since these areas are distributed throughout the material, they contribute to an increase in mechanical characteristics and prevent crack growth, according to [69,70,71]. The presence of chromium is noted both in molybdenum-enriched areas and evenly in the nickel-aluminum matrix, most likely in the form of a solid solution.

The distribution of all elements over their phase-space looks uniform. Nevertheless, according to the distribution maps of chromium and molybdenum in the NiAl matrix, it can be concluded that it contains much more chromium than molybdenum. Chromium is distributed between two phases, which is consistent with the results obtained in [72,73].

Figure 8 shows the distribution of elements in a composite sample reinforced with zirconium oxide nanoparticles. The phase regions of NiAl and CrMo are clearly observed. The CrMo phase is also present in the form of finely dispersed irregularly shaped inclusions along the grain boundaries.

The presence of calcium oxide is explained that the NiAl powder (PN70Yu30) and chromium powder (PH1S) may content ~0.1 wt.% Ca according to the manufacturer.

### 3.5. Bending Strength

Figure 9 shows the results of measuring the bending strength at 25 and 700 °C of composite samples modified with zirconium oxide and spinel nanoparticles obtained in this work compared to date for NiAl-Mo materials from other researchers [74,75]. The maximum strength of the studied in this work materials (475 MPa) at 700 °C corresponds to the sample reinforced by zirconium oxide nanoparticles in the amount of 0.1 wt.%.

A clear dynamic of the growth in bending strength was observed for samples with 0.05 and 0.1 wt.% of ZrO_2_ nanoparticles and 0.2 wt.% of MgAl_2_O_4_ nanoparticles with an increase in their concentration and test temperature. Samples with 0.01 wt.% of ZrO_2_, 0.01 wt.% and 0.1 wt.% of MgAl_2_O_4_ nanoparticles have an extreme behavior of the strength change on temperature increase. An interesting fact is that the addition of zirconium oxide or spinel nanoparticles significantly increases the high-temperature strength of the material. At higher additions, the tendency to aggregation is inherent for nanoparticles, and it is more difficult to distribute them uniformly, so it may decrease the matrix mechanical properties. It can show up in character of the properties-nanoparticles percent dependence curve as the presence of several extremums.

Young’s modulus was determined by the ultrasonic method (Figure 10). The average Young’s modulus of M was 205 GPa. The porosity of composites was in the range of 2.9–4.4%.

It seems to show that a change in the Young’s modulus-nanoparticles concentration curve of composites reinforced with spinel and zirconium oxide nanoparticles has extremums. The values of Young’s modulus are somewhat higher for the composite strengthened with 0.05 wt.% ZrO_2_ and 0.2 wt.% MgAl_2_O_4_ nanoparticles additions.

### 3.6. Microstructure of Material Fractures

Figure 11 shows the fracture microstructure of sintered samples of the M system. The typical pure M structure is shown in Figure 11a. The matrix grains size ranges from 5 to 25 μm. Chromium-molybdenum phase particles are located in chains along the boundaries of the matrix, that can be seen in Figure 11a,b. The size of the CrMo phase areas ranges from a few to tens of μm. The matrix fractured surface particulate qualities can be found in Figure 11c,d with cracks and grooves.

Figure 12 shows the cluster or clusters (aggregates) of aluminum-magnesium spinel nanoparticles and the resulting crack in the sample M-0.01 wt.% of MgAl_2_O_4_ after bending tests. As was shown in [75], the reasons for the strengthening of materials based on NiAl couldn’t be explained by any single effect. In relation to the materials obtained in this work, such reasons include: dispersion strengthening by microparticles of the CrMo phase, dispersion strengthening due to the addition of small amounts of nanoparticles of refractory compounds (ZrO_2_, MgAl_2_O_4_) [75]. In addition, the cause of strengthening is crack bending due to the creation of obstacles by the second phase, as can be seen in Figure 12. At the same time, it is important to remember that the difference between the elastic moduli of molybdenum (324 GPa) [76], chromium (279 GPa) and nickel-aluminum (188 GPa) [77] leads to the residual stresses increase around fine particles of the eutectic phase due to external impact [78]. This effect helps to limit the growth of cracks.

Figure 13 shows fracture images of composites with ZrO_2_ nanoparticles after bending tests at different temperatures. A hybrid nature of fractures is observed. There is intercrystalline fracture, which is confirmed by clearly visible crystal grains of the matrix, similar to chips. 

The crack is spreading on the surface of the grain faces and along the interphase surface (between the matrix and the eutectic phase). The destruction is also brittle that wavy microrelief indicates, i.e., the presence of chip facets with steps in the form of a wavy pattern. A crack usually grows along a crystallographic plane with small indices [79]. The chip planes have different directions, which is associated with the corresponding orientation of individual crystal grains. This is due to the characteristic type of fracture in the fractographic analysis of the microstructure of the material (Figure 13c) [79,80,81]. Steps of wavy fracture are typically the result of cleavage along second-order chip planes. They combine and spread in the direction of the crack to minimize energy. This can also be explained as capturing cracks by regions with higher fracture toughness, in this case it is CrMo [80] as in Figure 12. It is possible to determine the trajectory of crack propagation along the direction of wavy patterns [81]. Intercrystalline failure is observed in the areas of molybdenum particles clusters. The nature of the fractures does not change with an increase in the test temperature to 700 °C. Micropores are visible in many grains.

### 3.7. Results of Internal Friction Investigation

Figure 14 shows curves of internal friction depending on temperature (25–700 °C). Peaks are visible in the temperature ranges of 150–200 °C and 350–400 °C. The presence of these peaks is typical for a nickel-aluminum alloy, they are clearly visible in almost all diagrams. Relaxation occurs at 420–450 °C, and the peaks at these temperatures are most likely associated with grain boundary sliding (Ke peaks), as well as with dislocation motion. The decline after the peak at about 400 °C may be associated with the deceleration of this motion by inclusions in the composites.

In general, the curves of internal friction show a high phase stability of materials with an M matrix with the addition of nanoparticles in the considered test temperature range.

## 4. Discussion

According to [39], phase decomposition is observed at 1173 K for the Cr-Mo alloy with 44.2 at.% Mo. Continuous solutions in the liquid and solid state and the absence of any intermediate phases are typical for this system. A miscibility gap occurs below 880 °C in the solid state [39]. The oxidation resistance of Cr-Mo alloys rapidly decreases with increasing Mo content at the temperature about of 1000 °C [40,41], so its amount should not be too large. Based on the results of work [42], EDS and XRD methods show well the content of various phases formed in the studied alloys. The authors note that the NiAl-40Cr alloy consists mainly of two BCC phases, NiAl (B2) and Cr (A2). The addition of Mo leads to the formation of two intermetallic phases (AlMo_3_ and MoNi). AlMo_3_ seems to disappear at high Mo additions (over 20% of Mo), while MoNi is retained at even higher Mo concentrations. The NiAl-40Mo alloy practically contains the Mo and NiAl phases [42].

In [43], as a result of sintering the NiAl-28Cr-5.5Mo-0.5Zr powder by spark plasma method, the phase composition of the obtained material included NiAl phases and Cr (Mo) phases. The alloy obtained by electric arc melting of NiAl-20Cr-20Mo alloy contained BCC-B2 NiAl, BCC-Cr, AlMo_3_, NiMo (orthorombic), Me_x_O_y_ phases [42].

In this work, the main phases of the material are NiAl and CrMo after spark plasma sintering of the NiAl-13Cr-13Mo powder mixture at 1300 °C. These phase formations are also predicted by thermodynamic modeling in the Terra^®^ software. Free carbon from molds and graphite paper, which diffusion is due to electrotransfer, for example, can lead to the formation of chromium and molybdenum carbides in the sintered material.

A strengthening effect with increasing test temperature for materials with nanoparticles additions (Figure 9) can be associated both with homogenization (same as solution heat treatment), and additional dissolution of chromium or molybdenum in NiAl and partial ordering of the BCC phase, and with the resistance of strengthening phases to the dislocations motion. The homogenization of materials may occur due to slow heating via bending testing (10°/min in 60 min). Moreover, it is well known that NiAl has a brittle to ductile transition at 300–400 °C [1,82]. In addition, it is known for a number of brittle materials [83,84], that with an increase in the temperature of bending tests, an increase in the strength of the test sample occurs as a result of plastic effects. This is because the deformation leads to the elimination of residual pores and the healing of the formed cracks, which increases the bending strength of the material.

## 5. Conclusions

The calculations of the equilibrium phase composition of the NiAl-Cr-Mo-C system is carried out at a temperature range of 25–1327 °C and a pressure of 0.1 MPa. The main components presented in the system are NiAl and solid solution of CrMo. Via thermodynamic modelling, the existence of carbides Cr_7_C_3_ and Mo_2_C are also found. There is a change in the composition of the Cr-Mo solid solution with an increase in temperature, with an increase in chromium and a decrease in molybdenum amount, as thermochemistry calculations have shown.

Samples of new composites containing of MgAl_2_O_4_ or ZrO_2_ nanoparticles were obtained by spark plasma sintering of mechanically activated matrix powder. The main phases of the materials are NiAl and CrMo.

The ultimate bending strength of the obtained materials versus the test temperature relation is studied. The maximum strength at 700 °C (475 MPa) is found for material with zirconium oxide nanoparticles addition in the amount of 0.1 wt.%.

The fractures of materials have a hybrid nature: there is intercrystalline failure, which is confirmed by the fact that grains of the matrix, similar to chips, are clearly visible. The destruction also has a fragile nature, which is determined by the wavy microrelief.

In the study of internal friction, typical peaks of a nickel-aluminum alloy are found in the temperature ranges of 150–200 °C and 350–400 °C. Relaxation occurs at 420–450 °C, and the peaks at these temperatures are most likely associated with grain boundary sliding (Ke peaks), as well as with dislocations motion. The decrease in the peak after 400 °C may be associated with the inhibition of this movement by inclusions in the composites. The decline in the peak after 400 °C may be associated with the deceleration of the motion due to inclusions in the composites.

## Figures and Tables

**Figure 1 materials-15-05822-f001:**
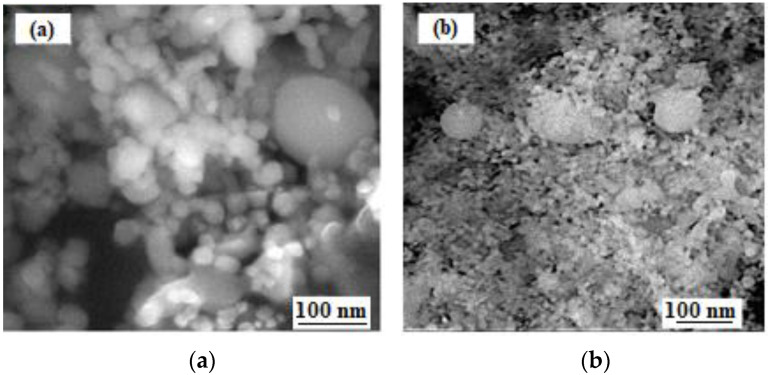
SEM images in the secondary electrons of zirconium oxide (**a**) and aluminum-magnesium spinel (**b**) nanopowders.

**Figure 2 materials-15-05822-f002:**
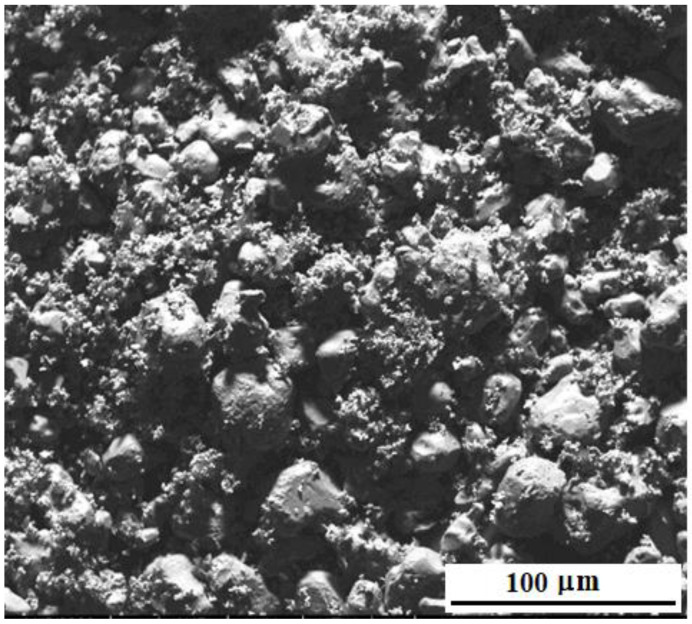
Microstructure of “M” powders after mixing in a ball mill.

**Figure 3 materials-15-05822-f003:**
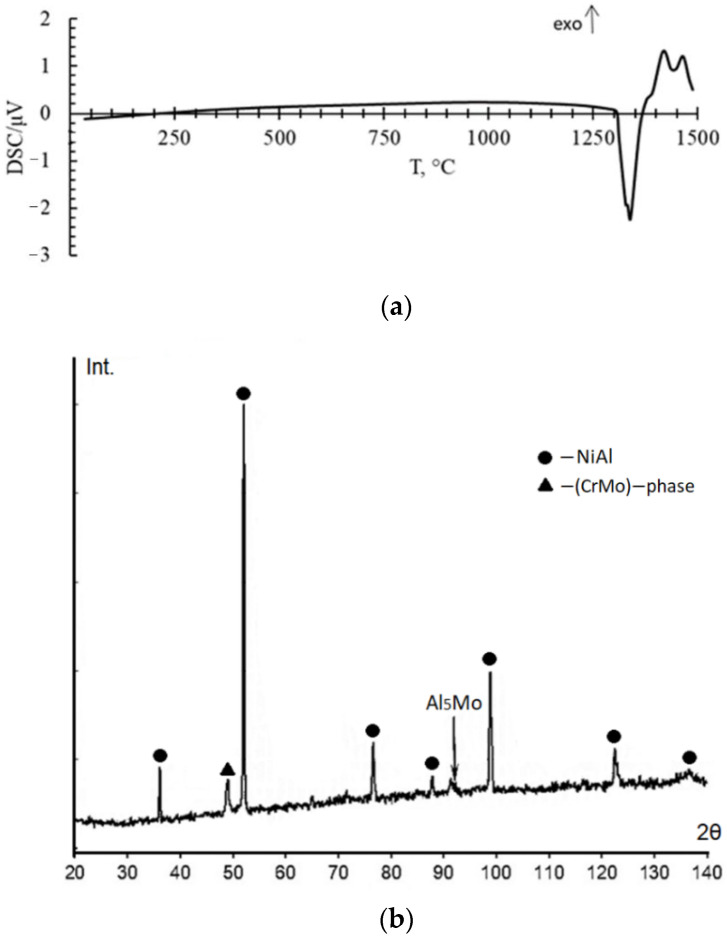
Results of DSC analysis of the matrix powders (**a**) and XRD spectrum of matrix samples after DSC (**b**).

**Figure 4 materials-15-05822-f004:**
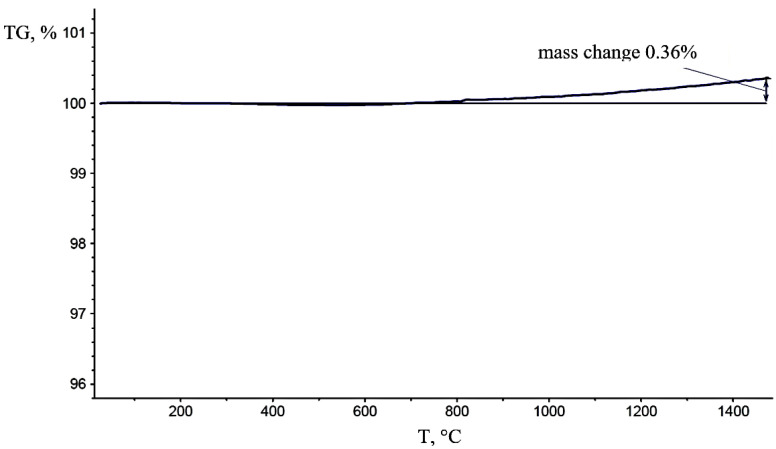
Change in the mass of the “M” composites powder.

**Figure 5 materials-15-05822-f005:**
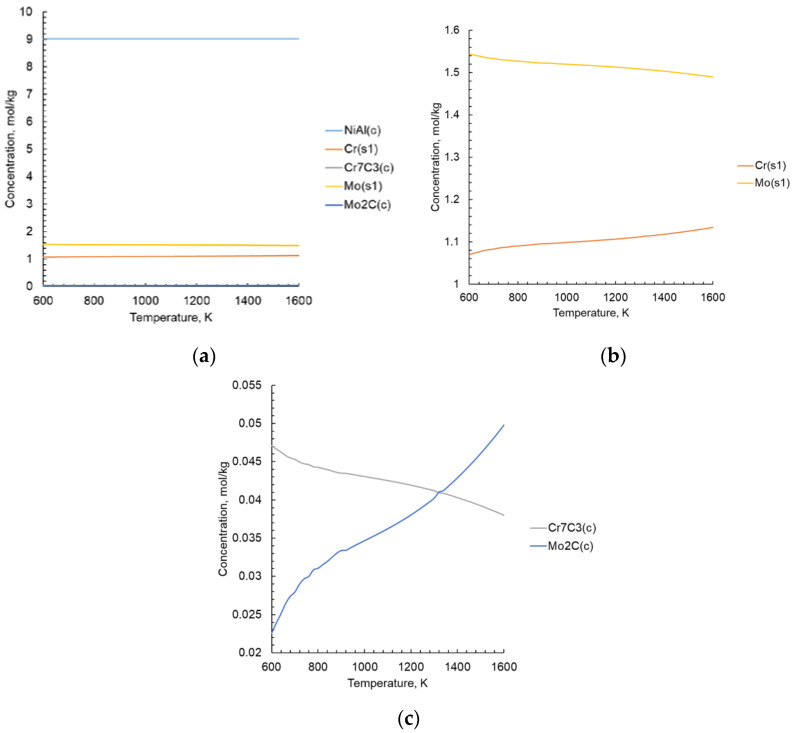
Results of calculating the equilibrium phase composition in the NiAl-Cr-Mo-C system: (**a**) all phases, (**b**) solid solution (s1) components, (**c**) carbide phases.

**Figure 6 materials-15-05822-f006:**
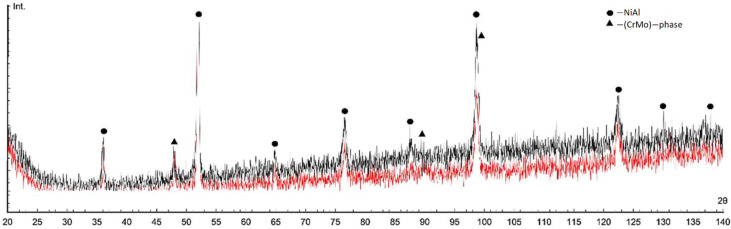
X-ray diffraction patterns of sintered matrix samples with 0.01 wt.% MgAl_2_O_4_ (black) or 0.01 wt.% ZrO_2_ (red) nanoparticles additions (Int.—intensity in arbitrary units).

**Figure 7 materials-15-05822-f007:**
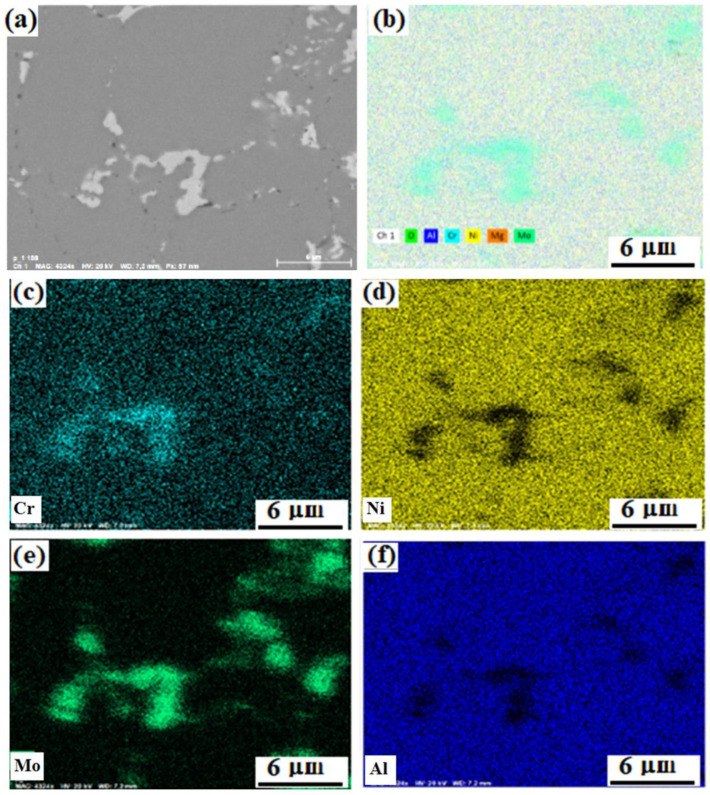
Distribution of elements in the structure of the composite M-0.01 wt.% of MgAl_2_O_4_: SEM backscattered electron image of the study region (**a**), summary map of the elements distribution (**b**), distribution maps of Cr (**c**), Ni (**d**), Mo (**e**), Al (**f**).

**Figure 8 materials-15-05822-f008:**
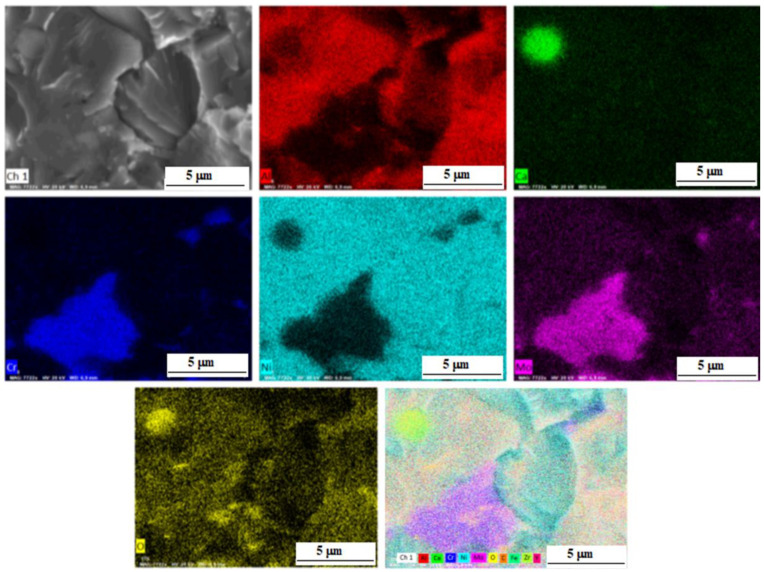
SEM secondary electron image, maps of the elements distribution in the sample M-0.05 wt.% ZrO_2_ and a summary map.

**Figure 9 materials-15-05822-f009:**
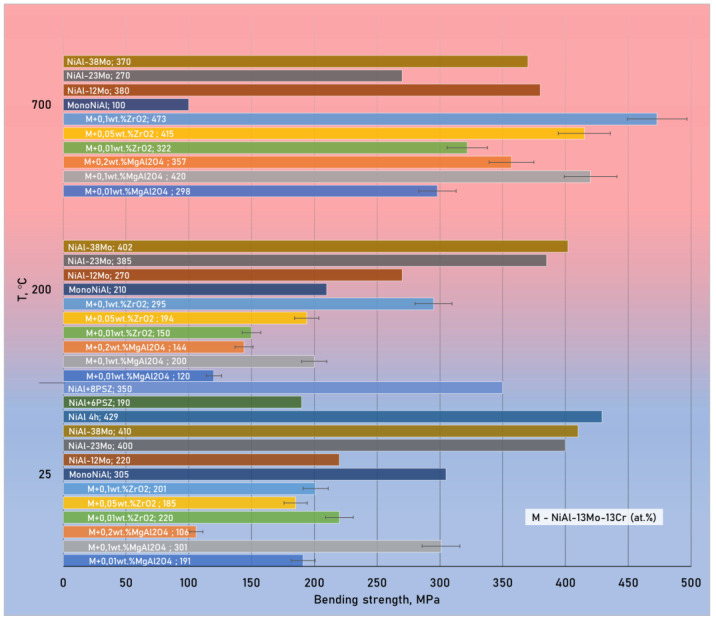
Change in bending strength of various composites versus test temperature (values for materials with the letter “M” were obtained in this investigation; other data are taken from [74,75]).

**Figure 10 materials-15-05822-f010:**
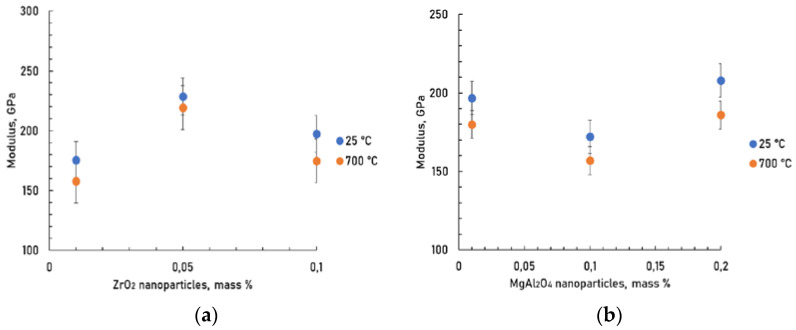
Young’s modulus of composites at test temperatures of 25 and 700 °C depending on the concentration of ZrO_2_ (**a**) and MgAl_2_O_4_ (**b**) nanoparticles.

**Figure 11 materials-15-05822-f011:**
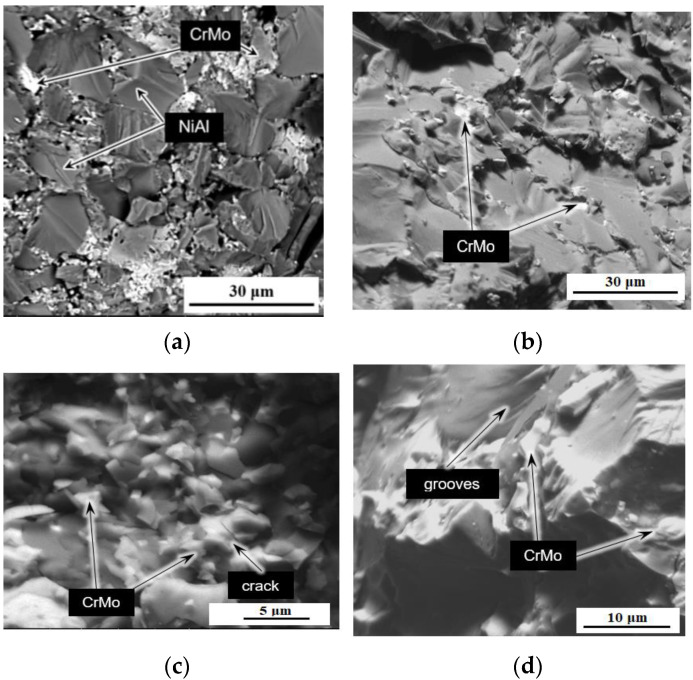
Microstructure of the sintered sample M without the addition of nanoparticles, SEM images in backscattered electrons: (**a**,**b**)—general views of the matrix microstructure, view of fractured matrix grains (**c**,**d**).

**Figure 12 materials-15-05822-f012:**
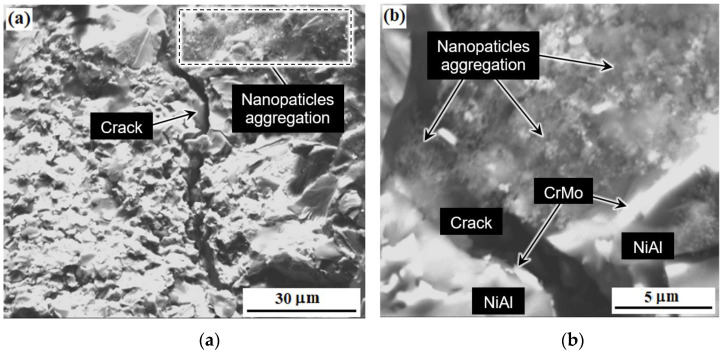
The microstructure of the material M-0.01 wt.% of MgAl_2_O_4_ after bending tests: microstructure of the formed crack area (**a**), clusters of aluminum-magnesium spinel nanoparticles (**b**).

**Figure 13 materials-15-05822-f013:**
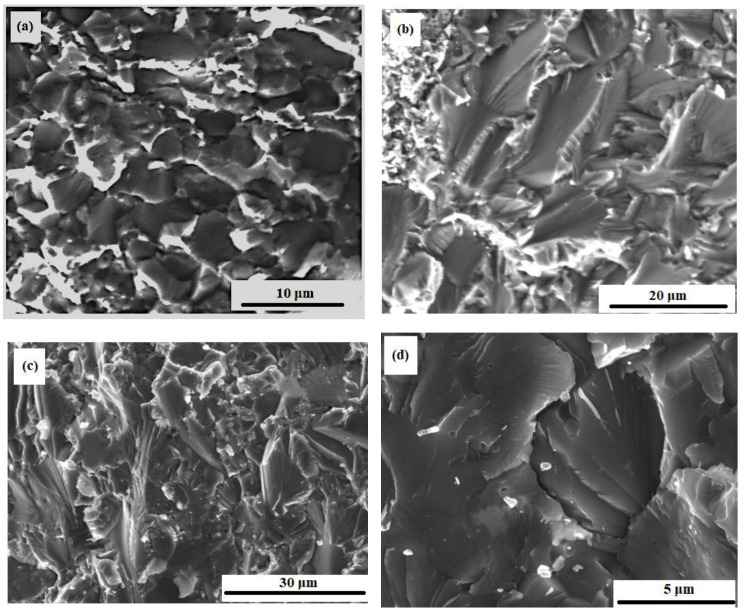
Fracture microstructure of the M-0.05 wt.% of ZrO_2_ sample after bending tests at: 20 °C (**a**), 200 °C (**b**), and 700 °C (**c**,**d**).

**Figure 14 materials-15-05822-f014:**
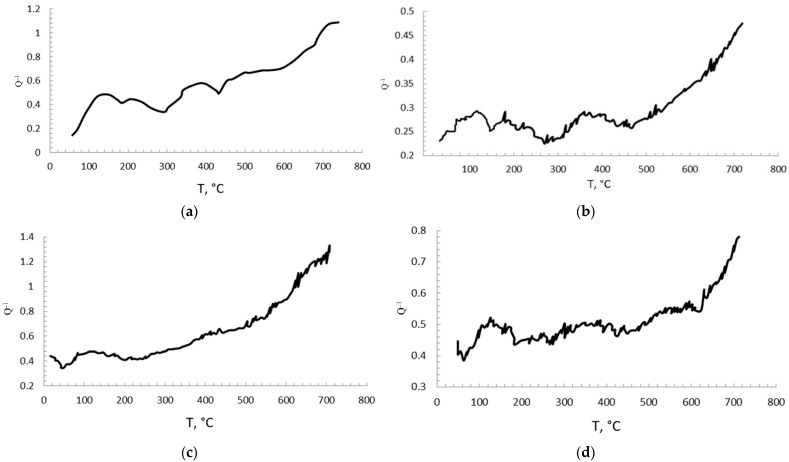
Damping curves of composites: M + 0.01 MgAl_2_O_4_ (**a**), M + 0.2 MgAl_2_O_4_ (**b**), M + 0.01 ZrO_2_ (**c**), M + 0.1 ZrO_2_ (**d**).

## Data Availability

Not applicable.

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
