# Peer review of "Preparation and Study of Composite Materials of the NiAl-Cr-Mo-Nanoparticles (ZrO2, MgAl2O4) System"

_materials, 2022, doi:10.3390/ma15175822_

Round 1

Reviewer 1 Report

This investigation account on the formation by spark plasma sintering and the 63 study of materials based on the system (at.%) NiAl-13Cr-with small 64 additions of zirconium oxide (0.01 - 0.1 wt.%) or aluminum magnesium spinel (0.01 - 0.2 65 wt.%) nanoparticles. The manuscript is not well structured format. Results are not presented and explained in proper manner.

Majors issues

1)    The novelty and the significance of the work is not well stated in the Introduction. The authors have presented a summary of the activities reported in the paper but is not clear for the reader how this work and the expected results are different from was published before.

2)    Scientific results are like a commodity for sale and therefore success in selling them depends on how you present your goods. The curves or graphs are not presented with great care a proof that the authors do not trust these results themselves.

So I could not recommend its publication in this form.

Author Response

Dear reviewer! Thank you for your comments.

Based on your comments, the following adjustments have been made, which are highlighted in yellow.

  1. Changes have been made to the relevant section with references to the literature.
  2. Reliability is ensured by a large amount of experimental material obtained using modern equipment, reliable and independent research methods, including electron microscopy, X-ray microanalysis, X-ray structural analysis, thermal analysis, etc., comparison and agreement of experimental results with literature data obtained under comparable conditions, and involving proven theoretical models and process mechanisms. All corrections have been added.

Reviewer 2 Report

Materials based on the NiAl-Cr-Mo system with zirconium oxide or aluminum-magnesium spinel nanoparticles additions are obtained by spark plasma sintering. The main phases are NiAl (B2) and CrMo. Bending strength measurements at different temperatures shows that nanoparticles of insoluble additives contribute to an increase in bending strength. A fractographic analysis of the samples fractures shows a hybrid nature of fractures and intercrystalline fracture. The results are useful. However, several limitations need to be overcome.

1.      The format of Fig.3, Fig.4,Fig.5,Fig.6 ,Fig.10 and Fig.14 should be improved, it is suggested to use a closed frame.

2.      TEM image of sintered sample M should be given.

3.      Fig. 11 is TEM or SEM image, it is should be given.

4.      Detailed comments should be given in Fig.11, for example, (a), (b) and(b).

5.      English writing needs further improvement.

Author Response

Dear reviewer! Thank you for your comments.

Based on your comments, the following adjustments have been made, which are highlighted in violet.

  1. Figures were corrected.
  2. We have added SEM image of the general view microstructure (Figure 11, a).
  3. This is a SEM image. Information has been added
  4. Description of figure 11 has been added to the text
  5. English has been improved.

Reviewer 3 Report

This study investigated the effect of nanoparticles on the microstructure and mechanical properties of NiAl-Cr-2Mo alloy. It is good to find that the particles ZrO2 and MgAl2O4 have an enormous potential to strengthen the NiAl-Cr- 2Mo alloy. However, there are also many problems that need to be further clarified in this study. The detailed comments are listed as follows:

1.     What is the size of the particles before and after milling? Please provide the data in the experimental procedure.

2.     The abbreviations should be explained when they first appeared in the study. For example, what is the meaning of P:B and P:L?

3.     In Fig. 7 and Fig. 8, the homogeneous colorful distribution of elements Mg and Zr should be caused by noise. The initial size of particles ZrO2 and MgAl2O4 seems very big in Fig. 1, and it is impossible to dissolve into the matrix completely during sintering. But in Fig. 7 and Fig.8, I can not find the particles. I am wondering the particles are not included in the areas shown in Fig. 7 and Fig. 8.

4.     Usually, the materials are softer at high temperatures, but in this study, the authors find a higher bending strength of the composites at high temperatures. The phenomenon is very interesting, but it lacks a reasonable and sufficient explanation. I strongly recommend a detailed discussion about this phenomenon 

Author Response

Dear reviewer! Thank you for your comments. 

Based on your comments, the following adjustments have been made, which are highlighted in green.

  1. Information about particle sizes before and after mixing was added.
  2. The abbreviations have been explained.
  3. The error in Figure 1 has been corrected ('nm' instead of 'µm'). Oxide particles do not dissolve in the matrix. The selected mapping scale is aimed at analyzing the distribution of the main components of the material, NiAl-CrMo. The Mg and Zr distribution maps have been deleted. Figure 12b shows nanoparticles aggregates.
  4. Explanation of strength increasing effect with temperature and references to literature were added.

Reviewer 4 Report

In your work "Preparation and study of composite materials of the NiAl-Cr-2 Mo-nanoparticles (ZrO2, MgAl2O4) system" you have described the preparation of starting powder mixtures, consolidation and properties of the above-mentioned materials. The work is interesting, however, some things need to be added/additionally explained before publishing.  

- what is the size of ZrO2 and MgAl2O4 powders (surface area is not so relevant here). 

- you have stated that you used sieving also to remove contaminants - what kind of contaminants? 

- I know that P:B and P:L  are familiar to a lot of readers, but the abbreviations need to be explained in the text

- heating rate during SPS is missing 

- study of the CHARGE?  What kind of charge? Check/correct also in the text... 

- row 124: Molybdenum particles are spherical.. - where do you observe this? Because from Figure 2 this can not be observed. 

- Figure 3 - the figure marks (a) and (b) are positioned bellow the image - correct this. What are the units in Figure 3b? Which XRD is presented in Fig.3b? 

- row 178: after DSC.. - clarify this 

- Figure 6: units and axis titles are missing. The visualisation of both XRDs would be clearer if, i.e., diffraction patterns would be stacked.

- row 185: the latter could not be identified - clarify, please 

- XRDs of sintered samples are missing

General remark: the term "nanoparticles" is used throughout the text, but the particles' size is not stated (you can not say that you have nanoparticles solely from Figure 1 - the magnification is low). 

- Fig 7.a - needs to be improved  - the image is not in focus

- row 215 : The EDXS mapping is done on fractured surface - not the most accurate. If possible do the phase maps on the polished surface. 

Also, statement that Ca-oxide is an artefact caused by sandblasting can not be used since the analysis was done on the fractured surface... 

From the presented results no indications of the position of  ZrO2 and  MgAl2O4 particles in the microstructure is evident. 

- Rows 223-229: Elaborate on this. What happens at higher additions... 

- the data for Young's modulus for "M" are missing 

- Figure 11: the description in the text is missing

- row 272: ... individual crystal grains.  - > how do you know this? Do you have a proof? 

- rows 273-279: Where is the crack upon which the discussion is based? 

- Results on internal friction: figure b - the measurements start above 100 °C

- row 322: how did you determine that you have carbides in your system? 

- row 323: how did you determine the change in the Cr-Mo composition? 

Author Response

Dear reviewer! Thank you for your comments.

Based on your comments, the following adjustments have been made, which are highlighted in turquoise.

- what is the size of ZrO2 and MgAl2O4 powders (surface area is not so relevant here).  – information about the average diameter of nanoparticles was added

- you have stated that you used sieving also to remove contaminants - what kind of contaminants?  - it meant large foreign inclusions which were not identified. We deleted this phrase.

- I know that P:B and P:L  are familiar to a lot of readers, but the abbreviations need to be explained in the text – The abbreviations have been explained

- heating rate during SPS is missing  - information was added

- study of the CHARGE?  What kind of charge? Check/correct also in the text... – This word has been changed to ‘matrix powder’

- row 124: Molybdenum particles are spherical.. - where do you observe this? Because from Figure 2 this can not be observed. – This redundant speculation was removed

- Figure 3 - the figure marks (a) and (b) are positioned bellow the image - correct this. What are the units in Figure 3b? Which XRD is presented in Fig.3b? – The positions were added and the units were corrected.

- row 178: after DSC.. - clarify this  - This information refers to sintered specimens (not after DSC analysis). We have corrected it and removed the error.

- Figure 6: units and axis titles are missing. The visualisation of both XRDs would be clearer if, i.e., diffraction patterns would be stacked. – The units and axis titles were added. These positions of the spectra correspond to real intensities of peaks measured in the same conditions.

- row 185: the latter could not be identified - clarify, please  - This word was removed

- XRDs of sintered samples are missing – This spectra  is in  Figure 6. The error was corrected

General remark: the term "nanoparticles" is used throughout the text, but the particles' size is not stated (you can not say that you have nanoparticles solely from Figure 1 - the magnification is low). – Error in scaling was corrected in figure 1 ('nm' instead of 'µm').

- Fig 7.a - needs to be improved  - the image is not in focus – Fig 7 has been improved.

- row 215 : The EDXS mapping is done on fractured surface - not the most accurate. If possible do the phase maps on the polished surface.  –Mapping has been done on fractured surface to avoid contamination of the samples during grinding and polishing.

Also, statement that Ca-oxide is an artefact caused by sandblasting can not be used since the analysis was done on the fractured surface... –This is due to the presence of calcium impurities, judging by the data from the manufacturer of NiAl and chromium powders. The error has been corrected,

From the presented results no indications of the position of  ZrO2 and  MgAl2O4 particles in the microstructure is evident.  – They are predominantly located along the grain boundaries [31,34,71]

- Rows 223-229: Elaborate on this. What happens at higher additions... – The information has been clarified, the text has been added to the section. With an increase in the concentration of nanoparticles, the risk of the complexity of their uniform distribution in the matrix increases due to the tendency to aggregation.

- the data for Young's modulus for "M" are missing  - Matrix Young's modulus data were added to the text

.

- Figure 11: the description in the text is missing – The description has been added for each figure.

- row 272: ... individual crystal grains.  - > how do you know this? Do you have a proof?  - This information has been updated with reference to the literature.

- rows 273-279: Where is the crack upon which the discussion is based?  - The crack is shown in figure 12. Descriptions were added in fig. 12.

- Results on internal friction: figure b - the measurements start above 100 °C – The figure was corrected.

- row 322: how did you determine that you have carbides in your system?  - The presence of carbides was determined by thermodynamic modeling, taking into account the uptake of carbon from molds during sintering into the samples. Fig. 5 a.

- row 323: how did you determine the change in the Cr-Mo composition?  - The change in the Cr-Mo composition was determined by thermodynamic simulation, Figure 5 b

Round 2

Reviewer 1 Report

My suggestions have not been taken into consideration by the authors. I have provided some comments and suggestions to help the authors to improve the manuscript.

The authors declared that: ''Reliability is ensured by a large amount of experimental material obtained using modern equipment, reliable and independent research methods, including electron microscopy, X-ray microanalysis, X-ray structural analysis, thermal analysis, etc., comparison and agreement of experimental results with literature data obtained under comparable conditions, and involving proven theoretical models and process mechanisms.''

In fact, the problem is not the number of analyses performed but rather the way  these results are presented. There must be some consistency in the presentation of results.  

1) Regarding my comments on the results presentation, the authors have just framed the graphs and no modification has been performed on the graphs in order to amaliorate thier presenttion.

2) Fig. 1 is a result from your investigation? If yes why did you present such result in experimental part?? By the way, It would have been more interesting to compare the microscopy of these two materials at the same magnifications. 

3) What is Fig. 2 refer to??

4) The following graphs need to be improved as an example Fig.3,  Fig.3a and b? what is the relationship between the two graphs? 

5) The XRD is an analysis that allows to determine the phase and  structure of a material why is this result presented in the 3.2 Results of thermodynamic modeling????  

6) Fig.4, no comment has been made on these graphs, so what was the point of presenting this graph (Fig.4) ? 

7) The graphs of Fig.5 are poorly presented as suggestion for the improvement, please use the same charcteristics (size, length and width) 

8) In section 3.3. (XRD analysis ) Please use colors to differentiate the two superimposed curves. 

  1.  

Author Response

Dear reviewer!

Figures 1, 5, 6 have been corrected.

The title of figure 2 shows that this is a mixture of powders of the matrix material, an explanation was added to the text.

As shown in the explanation to these figures (3 a, b), which are in the text, phase transformations occur at certain temperatures, most likely associated with the formation of intermetallic compounds and early melting. Lines 152-160

Figure 4 comments  - lines 164-166.

Thank you!

Reviewer 2 Report

The mauscript can be accepted now.

Author Response

Thank you very much!

The manuscript was corrected.

Reviewer 3 Report

The manuscript has been correctly appropriately.  It can be published 

Author Response

(The authors gave the same response as above.)

Reviewer 4 Report

The manuscript has been revised according to the suggestions of reviewers. I recommend its publication. 

Author Response

(The authors gave the same response as above.)
